# Detecting Walking Challenges in Gait Patterns Using a Capacitive Sensor Floor and Recurrent Neural Networks

**DOI:** 10.3390/s21041086

**Published:** 2021-02-05

**Authors:** Raoul Hoffmann, Hanna Brodowski, Axel Steinhage, Marcin Grzegorzek

**Affiliations:** 1SensProtect GmbH, 85635 Höhenkirchen-Siegertsbrunn, Germany; axel.steinhage@future-shape.com; 2Institute of Medical Informatics, University of Lübeck, 23538 Lübeck, Germany; grzegorzek@imi.uni-luebeck.de; 3Institute of Health Sciences, Department of Physiotherapy, Pain and Exercise Research Lübeck (P.E.R.L.), University of Lübeck, 23538 Lübeck, Germany; hanna.brodowski@uni-luebeck.de; 4Geriatrics Research Group, Charité-Universitätsmedizin Berlin, 13347 Berlin, Germany

**Keywords:** gait patterns, gait analysis, machine learning, feature learning, time series analysis, recurrent neural network, artificial neural network, sensor floor, SensFloor, long short-term memory, unilateral heel-rise test, dual-task

## Abstract

Gait patterns are a result of the complex kinematics that enable human two-legged locomotion, and they can reveal a lot about a person’s state and health. Analysing them is useful for researchers to get new insights into the course of diseases, and for physicians to track the progress after healing from injuries. When a person walks and is interfered with in any way, the resulting disturbance can show up and be found in the gait patterns. This paper describes an experimental setup for capturing gait patterns with a capacitive sensor floor, which can detect the time and position of foot contacts on the floor. With this setup, a dataset was recorded where 42 participants walked over a sensor floor in different modes, inter alia, normal pace, closed eyes, and dual-task. A recurrent neural network based on Long Short-Term Memory units was trained and evaluated for the classification task of recognising the walking mode solely from the floor sensor data. Furthermore, participants were asked to do the Unilateral Heel-Rise Test, and their gait was recorded before and after doing the test. Another neural network instance was trained to predict the number of repetitions participants were able to do on the test. As the results of the classification tasks turned out to be promising, the combination of this sensor floor and the recurrent neural network architecture seems like a good system for further investigation leading to applications in health and care.

## 1. Introduction

Evaluating gait patterns is an essential resource in diagnosing neurological states or orthopedic problems. They play an important role in monitoring the development of diseases and controlling clinical and therapeutic decisions [1], and can be used to track the progress after healing from an injury. By identifying the properties of gait patterns that deviate from the norm values, it is possible to decide on a precisely targeted therapy as an intervention back towards a more normal gait. Gait patterns can be picked up with all kinds of sensors, by measuring motion as in accelerations and rotations, or tracking locations of limbs or footfalls. For some of that, one can use sensors that are below the floor and which deliver the time and location of steps—the specific sensor floor model that was used for this study, SensFloor^®^, achieves this by detecting changes in the electric capacitance on a grid of sensor fields. The sensor floor is easy to install and integrates well with the environment as it is hidden below common flooring types. SensFloor allows collecting the most intensively researched parameters in this field of gait analysis like cadence, step width and step length, or the timings of the stance and swing phases that are found in human gait [2]. These parameters can then be used to identify asymmetries or for comparing them with tables of normal parameters for a person’s physique. Alternatively, gait can be analysed by formulating a task of interest as a classification or regression problem and train a machine learning model, which is the approach we chose for this project. This is useful when working with sensors, where patterns can be found directly in the data stream without a detour of calculating intermediate, semantically meaningful or descriptive parameters or features. In a pilot study [3] it was shown that the data generated by the sensor floor is suitable to train an artificial neural network to recognise if persons have a high risk of falling, or not. The study described here evaluated if this sensor technology is also capable of recognising very subtle changes in walking patterns. For this, data from a young and healthy cohort was collected and examined, which was expected to show only very small differences in the gait patterns within the cohort, and also in test-retest for every individual person. To induce changes in the gait patterns, extra requirements were set for the participants to be fulfilled while walking, like putting on a blindfold or performing a dual-task like spelling backwards [4,5]. These tasks were chosen as it was expected that they generate the very small variations in the gait patterns which were needed for the system evaluation. The detection of these subtle changes is a step towards applying the method in the future to applications such as the detection of changes in gait caused by neurological or orthopaedic diseases. The main contribution of this paper is in the methodological setup, which is a combination of using the floor sensor hardware for recording gait patterns, processing the raw data with its unique properties, and the application of machine learning models for the analysis tasks on the specific kind of data that was gathered. A trained system that automatically delivers relevant hints or predicts parameters which are of high clinical interest would save a lot of time and work in everyday clinical practice as it could help in identifying those patients who may benefit from targeted interventions.

## 2. Related Work

### 2.1. Gait Patterns, Interventions and the Unilateral Heel-Rise Test

The study protocol included different speeds of walking as well as walking challenges. These conditions were derived from findings in the research of gait analysis, which suggest that these walking modes should influence individuals’ gait parameters. It is generally known that the stance phase and swing phase in human gait vary due to gait speed. Slow walking speed leads to an increase of double limb support, while fast walking speed increases single limb support [2]. Further, it has already been demonstrated in young healthy adults as well as in older adults that visual impairments lead to changes in spatial and temporal parameters [6,7], thus one of our conditions was to let participants walk with closed eyes. Moreover, dual-tasking is a common feature in activities of daily living, generally including a motor and cognitive task [8]. The combination of walking with a mental tracking task, like spelling backwards (the task we chose), is often used to evaluate the effect in spatiotemporal gait parameters in older and in younger adults [5,9,10,11]. Finally, power and endurance of the calf muscles, especially M. triceps surae, is essential for human gait, balance and for mobility in everyday activities [2]. The importance of calf muscle strength in static and dynamic balance has been identified in several studies [12,13,14]. A standard test of muscle strength is a manual muscle testing with the examiner providing the resistance. Due to the short lever of ankle plantar flexors (M. triceps surae) this technique could affect ceiling effects. For this, a standing Unilateral Heel-Rise test (UHR test) that uses body weight as the resistance has been substituted [15]. The UHR test was used in combination with sensor analysis before [16]. In the current study, adults without lower-limb lesions performed as many unilateral heel rise repetitions as possible. The UHR-test procedure and the criteria for application are based on previous examinations [15,17]. It has not been examined yet if and how muscle calf endurance and strength affect spatiotemporal parameters in younger healthy adults.

### 2.2. Sensors for Gait and Behaviour Analysis

Depending on what one is interested in, different sensors can be used to collect different aspects of gait patterns. The historically first sensor that was used in gait analysis as early as in the 19th century was the camera [18,19]. In a modern form, it is still in use as cameras are widely available, and algorithms to process images are ever advancing [20]. Camera images can either be processed directly by extracting limb positions and joint angles, or by learning features with machine learning algorithms. Cameras are also often used in a motion capture approach, with markers reflecting infrared light, which are tracked in three dimensions with a very high precision and sample rate. Adding a spatial dimension to the camera image is possible with RGB-D cameras (Red-Green-Blue-Depth), like the Microsoft Kinect^®^, which was for example used previously for gait analysis in a classification problem in patients with Parkinson’s disease [21,22] or Multiple Sclerosis [23]. Further research using Kinect sensor was conducted in gait analysis of children with ataxia [24] or cerebral palsy [25].

Another very common type of sensors used in gait analysis are Wearable Inertial Measurement Units (IMUs). They are easy to handle, and gather motion data such as linear and rotational accelerations, as well as absolute orientation in the room via magnetometers and barometers, all of that in three dimensions and with a very high precision. In gait recording setups, they get attached to the limbs of the person who is being recorded. By varying the number of IMUs and their position, one can easily focus the data gathering process on aspects of the gait for a certain purpose.

A different, very direct way of measuring the timing and positioning of feet on the floor is to put the sensing elements directly onto or into the floor. There are a few measuring techniques that are suitable for this. Typically these systems either use force or pressure sensors, or exploit the fact that humans, with their high share of water in the body, influence electric fields and the electric capacitance. A common system for the purpose of gait analysis is the model GAITRite^®^. This system is a pressure-sensitive sensor with a very high spatial and temporal resolution. It is extensively used in gait research, and its validity and reliability has been demonstrated in several papers [26,27,28,29,30]. Other sensor floor projects that rely on measuring force are for instance described in [31] using pressure sensors, and [32] with a piezoelectric polymer. A completely different class of floor sensors relies on the measurement of electric field properties instead of force or pressure. The first type of these floor sensors measures the impedance of electric field couplings for an array of sensor plates, which is also called near field imaging [33]. A second type measures the electric capacitance, like the ELSI^®^ Smart Floor by Mari Mils [34,35], or SensFloor^®^ by Future-Shape [3,36,37,38,39,40,41,42,43]. The latter one was used for the data acquisition here, and is described in more detail later-on.

Each class of sensors has their own unique advantages. On the one hand, the advantage of pressure-sensitive floors is that they directly deliver information about forces applied to the floor. Given a high enough spatial resolution, one can even get a distribution of pressure that the foot exerts on the floor, which is a useful gait parameter in itself and which can give hints about malpositioning of the feet or a disturbed foot roll-over behaviour. This is hard to achieve with contactless measurement systems that measure capacitances or perturbations of the electric field, as they actually react on a combination of area and distance to the sensing units, but not necessarily forces. The contactless and forceless sensors, on the other hand, can be used under nearly all kind of flooring layers, as long as they are not conductive. This is not the case for pressure sensors, which need to be used with either no flooring layer on top, or a soft one that propagates the forces adequately. Generally, sensors that measure electric properties are therefore suitable to be used in larger areas like whole rooms and building floors, and they can be built to be more robust. There is no mechanical wear and they are protected and shielded by a common flooring layer that are often themselves engineered to last up to several decades. Generally speaking, floor sensors are more unobtrusive than other sensors. The anonymous and privacy-conserving way of tracking people with a sensor floor, especially when compared to camera-based approaches opens for instance the possibility to track customers in stores [44]. When using cameras, the environment usually has to be controlled for lighting and obstacles in the line of sight, and there are often a lot of cables running, which makes them unsuitable for some use cases, especially for a day-to-day practical use in a doctor’s office or hospital. Furthermore, the use of cameras comes with highly sensitive video data of individuals which can hardly be anonymised. This introduces questions of data privacy that have to be handled appropriately. For the application of IMUs, sensors have to be attached to the patient, which may be experienced as burdensome or uncomfortable, and takes some time that might not be available in the hospital routines. However, both camera and IMU sensors are very convenient and useful for research, where there are less concerns of practicality and robustness of the setup. Pressure sensor floors are convenient to use, but take up a lot of space that cannot be used otherwise. Plus, they can easily get damaged, for example when rolling a very heavy hospital bed with wheels over them. Capacitance based sensor floors have the same level of convenience in daily use and do not take up space, but come at the cost of significant changes to the premises on installation, as they have to be put below the flooring. In the prospect of practical applications they seem to be the most promising sensor for many use cases, given that the necessary construction work at the time of installation is tolerable.

### 2.3. Recurrent Neural Networks for Time Series Analysis

Machine Learning Models, and more specifically Artificial Neural Networks are a class of algorithms that were successfully applied to a wide variety of classification and regression tasks. The different neural network architectures have in common that they are usually used in a Supervised Learning manner, meaning that they are presented with inputs and correct outputs (or targets) in a training phase. In the training phase, internal weights are adjusted to minimise the error between the network output and the true output. The weight adjusting is done by Backpropagation of Errors [45,46]. By this, the network builds up an internal model of the inputs to outputs relation. For time series data, as are produced by the sensor floor used in this project, Recurrent Neural Networks (RNNs) showed to be especially powerful [47]. Recurrent neural networks stand in contrast to feedforward neural networks as they include internal feedback connections which carry neuron activations from previously seen inputs and internal states. This enables recurrent neural networks to have a memory of past inputs and activations, which is the reason for them being very useful for learning on time series data. They show very good results for instance in audio analysis applications like speech recognition [48,49,50] or handwriting recognition [51]. A very common variant of RNNs is the Long Short-Term Memory (LSTM) model [46]. The LSTM model was designed by combining internal input, output and forget gates with adjustable weights. Giving a general view on the function of these elements of a LSTM cell, the input gate controls the amount of new information that is processed at a time step inside the LSTM unit, while the output gate controls the amount of information that is forwarded to subsequent layers of the network. The forget gate controls how much and which information is kept or dropped (forgotten) from the previous timesteps. LSTM networks can be trained by backpropagation of errors, and they are less prone to problems like vanishing gradients, which is often the case with more basic Recurrent Neural Network architectures. Its special structure enables a fine-grained self-control over which values to remember in the training phase in sequence learning. This is also the architecture that we used here, followed by some densely connected layers, or Multi-Layer Perceptron [52,53]. The Multi-Layer Perceptron is a structure of several layers of neurons between the input and output layer. All outputs of the neurons of one layer are connected to all inputs of the neurons of the next layer, which is why they are also called densely connected layers. These connections consist of weights that are adjusted in the training phase by backpropagation of errors. Multi-Layer Perceptrons can be used in combination with LSTM layers. Generally, the major strength of artificial neural networks in regression and classification problems is that they will find relevant features in the input patterns autonomously in the training phase. Furthermore, once a good architecture is found for a certain type of input signal, it can be re-used for similar problems by training a new instance with other target labels—taking the current study as an example, the same network architecture was trained for different walking modes as targets.

## 3. Methods

### 3.1. Capacitive Floor Sensor

The floor sensor used for the data acquisition in this project is called SensFloor^®^ and is developed and produced by the company Future-Shape GmbH [37,38]. This system is of the type that makes use of measurements of the electric capacitance. The sensors can be installed in all indoor environments under common flooring materials, which enables many types of applications. Most commonly, the system is used in elderly care facilities for fall detection [40], but also for ambient assisted living at home [42]. Various approaches exist to extract gait parameters from the sensor data, for instance with an automatic step detection algorithm [36]. The SensFloor base material is a three-layered composite, with a thin aluminium foil at the bottom, a polyester fleece in the middle with a height of 3 mm, and a thin top layer of polyester fleece which is metal-coated and thereby electrically conductive. The floor sensor is organised as a grid of independently operating modules. Each module has a microcontroller board in the center of it, which is connected to power supply lines and eight triangular sensor shapes. Both the power supply lines and sensor shapes are created by cutting the conductive top layer of the base material and removing intermediate material. By doing this, the top layer can work in a similar way as a printed circuit board. For covering the floor of a room, the modules are put next to each other, and the power supply lines of adjacent modules are electrically connected with textile stripes of the same material as the top layer.

The topological advantage of using a triangular grid for the sensor fields is that it doubles the number of sensitive areas as compared to a rectangular grid of the same edge length. Most rooms have their walls meeting at right angles, therefore it is most convenient to engineer the outline of the whole module to have a rectangular shape. This way, the rectangular modules can reasonably be placed next to each other, aligned to all the walls for covering the whole room. By choosing a triangular shape for the sensor fields inside the module, one can connect eight instead of four sensor fields to every microcontroller board, thereby making better use of the microcontroller capabilities and increasing the spatial resolution. The installation is typically powered by a single power supply unit delivering 12 V, which can be connected at any position along the edges of the room. A schematic of how SensFloor is composed, and a photo of a real arrangement of modules after installation is shown in Figure 1. To accommodate for peculiarities of the ground plan such as columns or non-rectangular corners, the modules can even be cut into better fitting shapes (as long as the microcontroller circuit board remains undamaged). SensFloor comes in three different standard shapes which translate into three different spatial resolutions. The Low Resolution type means every module is a rectangle of size 1 m × 0.5 m (area of 0.031 m
${}^{2}$
), resulting in a spatial resolution of 16 sensor triangles per square meter. High Resolution modules have a square shape of size 0.5 m × 0.5 m (area of 0.031 m
${}^{2}$
) or 32 sensors per square meter, and Gait Resolution is the highest resolution: 0.38 m × 0.38 m (area of 0.018 m
${}^{2}$
) with approximately 55 sensors per square meter. For special applications, other resolutions and shapes can be produced by cutting the top layer respectively. The microcontrollers measure the electric capacitance of the eight sensor fields that are connected to it. When someone steps on a sensor field, the foot acts as the second plate of the capacitor system, increasing the measured capacitance. If, incidentally, both feet touch the same sensor field, the capacitance is increased further as a result from the increased covered area. The capacitance measurement is done with a sample rate of 10 Hz. However, the system does not report the measurements in a fixed-rate mode. Instead, consecutive measurements of the capacitance of a sensor field are compared to each other, and only if they differ by a certain amount for at least one sensor field of a module, a sensor message is generated. This behaviour is called event-based and has the advantage that there is less excess information generated by the sensor system at a very early stage in the sensor data processing pipeline. Sensor Messages are sent out by the modules over radio on the Industrial, Scientific, and Medical Band (ISM) on 868 Mhz or 920 Mhz (depending on the region and jurisdiction). A central transceiver collects the wireless sensor messages in a connection-less mode. As the radio range is quite high for this indoor use case (approximately 20 m), there is usually only one single transceiver needed per room. For very large rooms, multiple transceivers can be used. Due to the event-based nature of the sensor system, the data rate depends more on the number of people on the floor and how active they are (walking or standing) than on the floor area of the sensors. This makes it possible to install the sensor in rooms or buildings up to several hundreds of square meters. In the scope of behaviour and gait analysis, SensFloor was previously used for identifying persons in a sensor fusion setup together with wearable accelerometers [39]. It was also shown that it is possible to discern if a cat or human walks over the floor [41]. Using a multi layer perceptron with a feature extraction preprocessing step made it possible to distinguish between humans with a low or high risk of falling due to unstable gait [3] or to (roughly) estimate a walking person’s age [43].

### 3.2. From Sensfloor Messages to the State of Electric Capacitances

The sensor floor is built up of modules that operate independently. Every module is attached to eight sensor fields on which the electric capacitance is measured (see Figure 1). If one or more of the field capacitances changes a certain amount compared to the previous measurement for that field, a message is sent out via radio. The message contains the unique id of the module, and the current capacitance values of all eight fields, indexed by *j*. On the receiver side, these messages appear as a stream of messages over time which can be recorded. The measurements of the capacitances that are communicated by one received message are valid until there is a new message received from the same module. For this reason, past messages need to be remembered until they are invalidated by a new message concerning the same sensor fields. This is done by introducing a data structure that is updated with every arriving sensor message, which is called the sensor state (of electric capacitances at time *t*) 
$S_{t}$
. From the module id that is contained in the message, and the ordered numbering of the sensor fields of a module, a new id can be generated, which uniquely identifies a single sensor field. This id is called field id 
$id_{f}$
. For every sensor field, their number being *n* in a sensor floor installation, there is one entry in the sensor state at any time. The entries contain the field id 
$id_{f}$
, the position of the field centroid 
$p_{f}=\left(p_{f,x},p_{f,y}\right)$
 in 2D coordinates on the floor, the time of the last measurement of this field 
$t_{f}$
 as the time since the start of the recording, and the electric capacitance 
$c_{f}\in \left[0,1\right]$
 that was measured within the measurement range. In Figure 1, it is shown where 
$p_{f}$
 is located in a SensFloor setup. When having the length of the module edge as 
$l_{M}$
, and approximating the sensor field as right-angled triangles, one can get the eight field centroid positions from the module center position 
$p_{M,i\in n}$
 (which is known at installation time) by adding/subtracting 
$\frac{2}{3}\left(\frac{1}{2}l_{M}\right)$
 and 
$\frac{1}{3}\left(\frac{1}{2}l_{M}\right)$
, respectively (centroid position in any right-angle triangle). At the start of any recording, the initial state 
$S_{t=0}$
 is initialised with capacitances 
$c_{f,t=0}$
 and measurement times 
$t_{f,t=0}$
 all zero.

$$\begin{array}{c} \mathrm{Sensor}\;\mathrm{State}\;S_{t}=\left\{\left\{id_{f,i},p_{f,i},t_{f,i},c_{f,i}\right\}|i\in \left[1\ldots n\right]\right\} \end{array}$$


For updating the previous state 
$S_{t-1}$
 to get the new state 
$S_{t}$
 when a new message 
$M_{t}$
 arrives at time *t*, it is convenient to handle a SensFloor message as a set of updates on individual sensorfields. The state is updated by replacing the capacitances and update times of the sensor fields that are part of the message.

$$\begin{array}{cc} \mathrm{Message}\;M_{t} & =\{t_{m},id_{mod},c_{m1},c_{m2},c_{m3},c_{m4},c_{m5},c_{m6},c_{m7},c_{m8}\}\\ \; & \equiv \{\left\{id_{f,j},p_{f,j},t_{m},c_{f,j}\right\}|j\in \left[1\ldots 8\right]\}\\ \mathrm{Updated}\;\mathrm{State}\;S_{t} & =\left(S_{t-1}\backslash \left\{\left\{id_{f},p_{f},t_{f},c_{f}\right\}\in S_{t-1}|id_{f}\in M_{t}\right\}\right)\cup M_{t} \end{array}$$


The sensor state 
$S_{t}$
 can be interpreted as a discrete-time dynamical system. In this analogy, the message 
$M_{t}$
 would be the only input to the system. The sensor state is similar to a camera image as it is a snapshot of the capacitances at one point in time. Processing the time series of these states makes it possible to track persons moving on the floor over time, or extracting foot positions for gait parameter calculation. Here, we directly use the time series of states as input to an artificial neural network after applying a geometric transformation and resampling.

### 3.3. Transformation to Local Coordinates

The sensor floor delivers data as a time series of capacitance measurements in a two-dimensional coordinate system that is aligned with the floor. The sensor can be used in arbitrary dimensions and shapes, following the ground plan of a building, and expanding to extremely large areas of floor. It is therefore not useful to take the sensor data and process it in the global x/y coordinates it is delivered in, as this would hardly generalise to other shapes and dimensions. Furthermore, the properties of gait are generally neither dependent on the location of where a person is walking, nor on the walking direction, they are therefore translation- and rotation-invariant. For these reasons it seems worthwile to only look at the sensor activations in the close vicinity of a walking person, as it includes all the information that is relevant and necessary to extract gait patterns and greatly reduces the input dimensionality by ignoring all the sensor fields that are too far away to measure associated footfalls anyway. The preprocessing step transforms the sensor activations as given in the sensor state of the sensor floor to the position and walking direction of the person, and removes all sensor activations that are too far away from the tracked position to be considered as relevant. This process is shown graphically in Figure 2.

In our recordings only the person to be recorded was supposed to be in the region of interest. In a real scenario, a more sophisticated tracker should be used which can also separate multiple persons walking on one floor area. In the case of only one person on the floor, their position 
$x_{p}$
 can be tracked by taking the weighted mean of the positions of all active sensor fields, with “active” meaning that the capacitance is above a certain threshold. Filtering by applying a threshold leads to the sparse representation of the sensor state 
$S_{s,t}$
. The threshold capacitance 
$c_{threshold}$
 can be chosen arbitrarily. In the current study, based on previous observations of sensor fluctuations, 
0.03
 (
3%
 of the measuring range) was chosen to account for noise. The position vectors of the remaining sensor fields in the sparse state are weighted (multiplied) with their capacitances, added up and normalised by the total sum of capacitances of the state. This approach is similar to calculating the center of gravity or center of mass of a particle system:
$$\begin{array}{cccc} \mathrm{Sparse}\;\mathrm{State} & \; & S_{s,t} & =\{\left\{p_{f},c_{f}\right\}\in S_{t}|c_{f}>c_{threshold}\}\\ \mathrm{Total}\;\mathrm{Capacitance} & \; & C_{t} & =\sum_{\left\{c_{f}\right\}\in S_{t}}c_{f}\\ \mathrm{Tracked}\;\mathrm{Position} & \; & x_{p,t} & =\frac{1}{C_{t}}\sum_{\left\{p_{f},c_{f}\right\}\in S_{t}}c_{f}p_{f} \end{array}$$


The walking direction is determined by taking the vector from the first to the last tracked position points, using the first and last state that were captured during a recording, respectively. From this, the average walking angle relative to the global *y* axis can be derived, which is used for rotating the sensor field positions into the local coordinate system of the walking person.

$$\begin{array}{cc} \mathrm{Walking}\;\mathrm{Direction}\;d & =x_{p,t_{end}}-x_{p,t_{start}},\;\hat{d}=\frac{d}{\left|d\right|}\\ \mathrm{Walking}\;\mathrm{Angle}\;\theta & =\arccos\hat{d}\cdot \left(\begin{array}{c} 0\\ 1 \end{array}\right) \end{array}$$


All entries in the sparse state that are further away from the tracked person’s position 
$x_{p,t}$
 than 
$r_{s}$
 are removed. Usually, this should not happen, but it can be the case if any capacitance reading exceeds the threshold due to higher than normal noise (e.g., if someone else than the recorded person accidentally steps into the recording area). Combining translation to the tracked position, rotation into the walking direction, and removing all field entries that are too far away, the sparse state in local coordinates then follows as:
$$\begin{array}{cc} \; & \mathrm{Local}\;\mathrm{Sparse}\;\mathrm{State}\\ \; & S_{l,s,t}=\{\{\left[\begin{array}{cc} \cos\theta & -\sin\theta \\ \sin\theta & \cos\theta \end{array}\right]\left(p_{f}-x_{p,t}\right),c_{f}\}|\left\{p_{f},c_{f}\right\}\in S_{s,t}\wedge |p_{f}-x_{p,t}\left|<r_{s}\right\} \end{array}$$


As a final step, the local sparse state field positions get resampled to a local 2D cartesian grid, with a positional resolution of 
$l_{s}$
 and again limited to a radius of 
$r_{s}$
. The grid points have a capacitance value associated. This is comparable to a receptive field for the artificial neural network.

$$\begin{array}{cccc} \mathrm{Local}\;\mathrm{Grid} & \; & G_{t=0} & =\{\left\{c_{G}=0,p_{G}=\left(\begin{array}{c} xl_{s}\\ yl_{s} \end{array}\right)\right\}|x,y\in Z\wedge \left|\left(\begin{array}{c} xl_{s}\\ yl_{s} \end{array}\right)\right|<r_{s}\} \end{array}$$


For every field that is contained in a local sparse state 
$S_{l,s,t}$
, the grid capacitance value 
$c_{G}$
 of the grid point with the position 
$p_{G}$
 that is closest to the field centroid 
$p_{f}$
 is updated to the corresponding value of 
$c_{f}$
. This resampling step is necessary as at some point, the field capacitances (at one timestep) are reshaped into a 1D vector which corresponds to the array of input neurons of the artificial neural network. It is done in a fixed association manner, any one grid position in the local coordinate system is therefore connected to the same input neuron. The whole transformation and resampling procedure is shown (with example data) in Figure 2. The task of associating input vector entry positions that are close to each other geometrically, and which therefore have a similar meaning for the gait pattern data analysis, is left to the artificial neural network. The capacitance value of the grid point is used as the activation value of the respective input neuron. Resampling the sensor activations into the local context of the participant’s position and walking direction results in a reduced set of positions and activations in the vicinity and local coordinate system. It is a sparse view on the whole state of sensor activations that contains all the relevant information for gait analysis. This set can be unrolled into a vector of activations by discarding the positions and collecting the capacitances in any order as shown in Figure 2 with the blue meandering arrow. By applying the state update, transformation and resampling at any time a message arrives, one gets a time series of vectors that is very suitable to use as input to time series machine learning algorithm like the Long Short-Term Memory based artificial neural network that we used here.

### 3.4. Data Collection

The data recordings were carried out in the APPS Lab (Assessment of Physiological and Psychological Signals Lab) at the Institute of Medical Informatics, University of Lübeck. In this lab, a room with a floor area of 24 m
${}^{2}$
 is fully equipped with the SensFloor capacitive sensor system. For this installation, the Gait Resolution variant of SensFloor was used, with a module side length of 0.38 m. The sensors cover the whole area of the room except for a negligible slim (10 cm) non-sensitive zone along the border which is needed for the power supply of the sensors. In Figure 1, it is shown what the system looks like before the final floor covering is installed on top. After completing the installation, the room looks just like any other normal room, as the sensor system is not visible any longer. In addition, Inertial Measurement Units (IMUs) delivered movement data for the gait pattern recording. For this, four devices of the LPMS-B2 Series from the Company LP-RESEARCH Inc. were used. This model is a combination of several sensors like an accelerometer, gyroscope, magnetometer and barometer, and can thereby produce a comprehensive measurement of the motion of the device in all three dimensions. The IMU data was sampled with a fixed rate of 50 Hz. Although the IMU data is not part of the current analysis, it will be in the focus of a later examination. Both the IMU and SensFloor data were collected and recorded at a central small-form-factor Intel NUC computer running Ubuntu 18.04 and ROS Melodic [54], the Robot Operating System on top.

By using ROS and its included data handling tools, it was ensured that all data was recorded in a time-synchronized manner although it was gathered from multiple sources. The IMUs had a Bluetooth connection to the recording computer and the SensFloor Data Transceiver transmitted data over an Ethernet network connection. Before the start of the recording, IMUs were attached to the participants in the following position: two at the outside facing side of the ankle, one at the lower abdomen (e.g., belt buckle height), and one at the sternum. As part of this project, a recording software and graphical interface was developed. The software ran on the recording computer, handled the connections to the sensors and the starting and stopping of recordings. Over a wireless network, it offered a graphical user interface running in the web browser to the person who supervised the recordings. In that interface, the current connection state of all sensors is shown, to inform the recording supervisor whether it is fine to proceed with a recording. A free text field is available for entering an ID for the participant. This ID is associated with the sensor recordings and also noted down on the case report form regarding that participant. The web interface also shows a recording ID, which increases automatically after every recording. The recording ID is noted down by the supervisor together with the walking interference mode that was performed by the participant for the respective recording. Finally, the interface has buttons for starting and stopping the recording, and resetting the connection to the sensors in case of any problems. Participants of the study started in the corridor outside the room, walked into the room, and in a straight line towards the windowsill opposing the door. Arriving there, they turned on spot, walked back and left the room again through the door. This procedure was repeated five times per participant and per walking mode. The start point was chosen to be outside the room such that the first few steps were not captured by the sensors and could thereby easily be disregarded. This is useful as when starting to walk, the gait patterns typically look different while accelerating from standing still, as compared to after taking some steps. The steps right before and right after the turning point inside the room which were captured by the sensor, were excluded for the same reason. The participants were asked to walk in one of five different modes in the following order: “walk at your normal pace”, “walk faster than normal”, “walk slower than normal”, “walk with your eyes closed”, “walk while spelling a word backwards (dual-task)”. For the dual-task mode the instructor told the participant a random word from a list of 100 words. After that, they performed the Unilateral Heel-Rise (UHR) test, and then had their walk recorded again, which we call “post UHR” recording. An example of a recorded walk is shown in Figure 3. For the dataset, 42 participants (age 19 to 31, M = 25, SD = 3.1) were recorded in total. Summed up over all walking modes, they performed 2506 walks from either door to window or window to door. All subjects gave their informed consent for inclusion before they participated in the study. The study was conducted in accordance with the Declaration of Helsinki, and the protocol was approved by the Ethics Committee of the University of Lübeck on 4 June 2020, file reference 20-214.

### 3.5. Data Analysis

All recordings were preprocessed by transforming the sensor activations into the local context, thereby building a time series of 1D vectors. The time series of every recording was further split into multiple time series of equal length by moving a time window of 30 steps over the whole recording in increments of one step. These were then used to train artificial neural networks. The architecture that turned out to work best for our case was a stack of one Long Short-Term Memory (LSTM) layer directly after the input layer, followed by four dense layers. The LSTM layer had an output size of 20, and the dense layers had 20 neurons each. The dense layer neurons were Rectified Linear Units (ReLU) with a linear identity activation function for input values above zero, and all zero output for input values below or equal to zero. Several goals exist that were tackled by solving classification and regression tasks. The output layer size was two neurons with Softmax activation function for the classification tasks, and one ReLU neuron for the regression tasks. As error metric and loss function the Mean Squared Error was chosen for the regression analysis tasks, and the binary cross-entropy for the classification task. The artificial neural network was trained with early stopping in the case of non-improvement on the validation error with a patience of 12 epochs, with a maximum number of 60 epochs for training. The implementation was done using the Tensorflow library (v2.3.0) in Python (v3.8.5) [55]. To assess the robustness of the model and the whole training and evaluation process, every run was done multiple times, to ensure that results were not incidental. Furthermore, the random seed for the network operations was not set to a fixed value, but chosen randomly every time. All networks were evaluated by Leave-One-Out-Crossvalidation (LOOCV). For the idiosyncratic analysis plan, one single walk was left out for the test set, one walk was left out for the validation set, and the rest was used for training, as the goal was to detect intrapersonal differences. For the generalised analysis, all recordings from one participant were left out from training for each test and validation set, as the goal was there to train the neural networks on patterns that were common to all participants. The split into an idiosyncratic and a generalised analysis was done to get results for both possibilities; (1) that the walking modes are reflected on a very individual level in the gait patterns, and (2) the case that there is a common effect on the gait patterns over the participants. In summary, neural networks were trained in three different experiment designs:

Predicting the Walking Mode—Idiosyncratic: In the first experiment, one artificial neural network was trained for every individual participant of the study and interference mode (except for the different walking speeds, as this was beyond the scope of this paper). The goal of this was to find out if there are intrapersonal differences in the walking patterns of the different modes that can be learned by the network. The task was set out as a binary classification task between the class of walking patterns from the normal mode and either “closed-eyes”, “dual-task” or “post UHR”.

Predicting the Walking Mode—Generalised: For this experiment, the machine learning model was trained on the walk data of all participants. The goal was to evaluate the possibility of classifying between the walk at normal pace and the interference modes, but this time in a generalised manner, to find out if there is a change in pattern that is common to all participants.

Predicting the UHR Repetitions: The goal of the last experiment was to predict the results a participant would achieve on the Unilateral Heel-Rise Test as a regression task. The neural network was again trained to achieve generalisation across participants. One network was trained for each leg, left and right.

## 4. Results

### 4.1. Results for Predicting the Walking Mode—Idiosyncratic

For the closed-eyes walking task, the classification accuracies for the participants on an intrapersonal, idiosyncratic level varied between 0.32 and 0.96, (M = 0.68, SD = 0.16). For the dual-task recordings, the range of accuracies was between 0.30 and 0.92, (M = 0.54, SD = 0.16), and for the post UHR walk between 0.15 and 0.87 (M = 0.52, SD = 0.15). The accuracy calculates as the amount of correct classifications divided by the number of all predictions, thus guessing corresponds to a value of 0.5. These results are graphically presented in Figure 4.

### 4.2. Results for Predicting the Walking Mode—Generalised

The network that was trained to generalise over walking patterns that are common to the whole cohort produced classification results that are shown in Table 1. Precision (also known as positive predictive value) calculates as the share of true positive predictions of all positive predictions, when the interference mode is the positive outcome. Recall (or sensitivity) is defined as the division of true positive predictions by all positive data points. 
$F_{1}$
 is the harmonic mean of Precision and Recall. The values for Precision, Recall and 
$F_{1}$
 Score happen to fall on the same value for the first two comparisons.

### 4.3. Results for Predicting the UHR Repetitions

The mean number of maximum heel rises performed in this cohort was 25.7 (SD = 7.8, with a minimum of 10 and a maximum of 45 repetitions) for the right leg and 24.3 (SD = 7.9, with a minimum of 13 and a maximum of 45 repetitions) for the left leg. When predicting the results on the UHR test from the “normal” gait recording, the artificial neural network performed with a Root Mean Square Error of 11.7. The true and predicted values did not correlate (Pearson coefficient of −0.02). The high error and low correlation is a limit of the current study.

## 5. Discussion

### 5.1. Summary

In this project, we aimed to evaluate the capability of SensFloor data for use with a recurrent neural network architecture to learn subtle differences in gait. The sensor floor that was used here delivers its data as a time series of capacitance measurements in two dimensions. For time series data, recurrent neural network architectures like Long Short-Term Memory Units (LSTM) often work especially well. This was also the case in this project, where we trained a LSTM-based neural network on the sensor data. For the dataset, we recorded participants in different walking modes. At first, the participants were told to walk at their normal pace, then faster than normal, followed by walking slower than normal. Then, the participants kept their eyes closed and their gait was recorded again. As another intervention, the participants carried out a dual-task walk, where they were asked to spell words backwards while walking and again being recorded. Subsequently, the participants performed the Unilateral Heel-Rise test (UHR), and were then again recorded walking at their normal pace and unimpeded. The intention of letting the participants walk at these different modes was to artificially introduce some variation into the gait patterns by interfering with their normal walking. Consequently, the first goal of training the neural networks was to have them learn the differences between the normal walking patterns and the artificially disturbed walking patterns or the walk immediately after the UHR test. The second goal was to predict the number of heel rise repetitions a participant could do. The UHR test is a feasible measure of muscle strength in the foot moving apparatus, and therefore a surrogate marker for the gait patterns. The results show that it is in principle possible to distinguish between the different walking modes. Overall, the idiosyncratic analysis, where the walking mode was classified after training the neural network intraindividually on the data of only one person, one after the other, showed the best results, while the generalised predictions were only satisfying for distinguishing between a normal and a closed-eyes walk. In this current study sensor floor was not able to reliably predict the number of repetitions in the UHR test.

### 5.2. Interpretation

As a first important result, we found that a neural network can distinguish between different walking modes, which works best when person-dependent analyses are used. For some persons, it worked very good (up to 95% accuracy), while it did not work at all for others (<50% accuracy, which is not better than guessing). We attribute this outcome to personal differences in walking and coordination in a way that for some persons, it just might not be too much of an interference to walk with closed eyes or walking while spelling words backwards. Speculatively, such differences could stem, for example, from age, athleticism, fatigue, or the ability to multitask. This would also explain the promising, but not perfect results for the generalised classification. When the walking pattern differences are really person-dependent, the artificial neural network will be presented lots of similar input examples with different target labels for the person who are not challenged by having to walk with closed eyes or dual-tasking. The generalised walking mode classification approach worked best for classifying between the closed-eyes and normal walk, and worse for the distinction between dual-task and normal walking mode. It seems like the closed-eyes mode is the one that introduces the highest disturbance to the gait coordination, resulting in the greatest variation to the walk patterns as compared to the normal walk. This is in line with previous work that shows that visual impairments also affect the gait of young and healthy adults [7].

It was not possible for the artificial neural network to learn differences between the walks before and after doing the UHR in the generalised classification, nor was it possible to predict the number of repetitions. This should not imply that there is no effect from doing the UHR, but it might be a limitation of the sensor, sample, and analysis that renders the distinction impossible, for example because they are too subtle to be captured by the relatively low floor sensor resolution.

### 5.3. Limitations

In general, some limitations arise from the choice of sensor floor. For gait analysis, e.g., the GaitRite system has a higher resolution in space as well as in time. So, if one only aims for the best quality in reaching the classification and regression goals, another sensor might work better. However, the SensFloor system is suitable for other areas of application. As it is very robust, it is reasonable to use it as a part of everyday medical routine. It was due to the prospect of actual usefulness in a clinical setting that we chose to check what information could be extracted from data generated by this kind of sensor. As participants do not really perceive the floor as a medical device or a piece of technology at all, the recording does not feel like an examination situation, thus we expect a more natural gait than in other settings.

The processing scheme that was chosen for the data analysis introduces some limitations as well. Although we aimed for a minimal loss of data by avoiding preprocessing steps as much as possible, a small positional error is introduced by resampling the floor sensor data into a grid in the local view of the walking participant, thereby discretising the field positions. However, the resulting time series of vectors is a perfect format as input for Machine Learning algorithms of all kinds. We chose this architecture as it is a good fit to the time series sensor data.

### 5.4. Implications and Outlook

The actuation of the calf muscles is a very important screw during the stance phase of human gait and the heel rise test is a common test to assess calf muscle endurance and strength. In a reliability study of the UHR test [56] with 40 healthy adults over 18 years and without current ankle injury or chronic ankle pain, the following results were declared as clinical reference: mean plantarflexion of 23 with a standard deviation of 13.3 repetitions. In our cohort, the mean number of heel rises was slightly higher, but comparable, with 25.7 (SD = 7.8, right leg) and 24.3 (SD = 6.9, left leg). Thus, we can be confident that our results are representative for clinical practice. Still, we analysed a young and healthy cohort with relatively high plantarflexion mean repetitions. We do not know yet, if there are no differences in gait parameters before and after the UHR test or if these differences are too low for the sensor floor to identify. It must be evaluated in further studies if this method could predict repetitions within a cohort of older adults, 65 years and above, with poorer functional abilities, lower muscle endurance of the M.triceps surae and more asymmetries in gait parameters. Furthermore, further research in a young healthy cohort is needed, using another test protocol like calf muscle training over a time of weeks or months, to enable the differentiation between a pre- and post-test using SensFloor data. Furthermore, it could be worthwile to examine if the gait patterns vary enough between the participants to be able to distinguish between individuals.

The dataset that was recorded also contains measurements from Inertial Measurement Units. This data was not yet evaluated and not part of this study. Other gait analysis studies [57] showed that a lot of information is contained in IMU data, especially about aspects that the floor sensor can not cover, like joint movements and angles. However, whenever integration is involved in the processing of IMU data, there are errors generated from drifts. This is not the case with the SensFloor, which has a global fixed reference frame for the capacitance measurements. Therefore, the sensor floor data is a perfect complement to the IMUs as it delivers its measurements in absolute positions which are not prone to drift over time. It appears to be very promising to work towards a sensor fusion approach combining the best properties of both sensors.

## 6. Conclusions

We gained insights into walking patterns of people with a sensor floor that measures the electric capacitance on a discrete grid of triangular sensor fields over time. For the dataset, we recorded participants in different walking modes which were expected to induce changes in the walking patterns. It was shown that the sensor data generally contains enough information for detecting walking challenges in gait with a Machine Learning approach. We aimed for a minimal loss of data in the processing chain by choosing a methodological approach of learning features directly from the raw stream following some minimal geometric transformations and resampling. The approach as it is described is a promising path, because it can easily be transferred to other applications and project goals in gait analysis using this sensor and analysis strategy. A very relevant future research goal that should be pursued in follow-up research this way concerns a cohort of elderly people, who are often prone to falls and where higher variances in gait can be expected as frailty is more prevalent. This age group could profit a lot from a better availability of gait analysis. The floor sensor is a viable implementation of a gait pattern recording sensor that is suitable for everyday practical use and thereby opens the possibility of a wide variety of applications. Given further developments of the proposed algorithms, the unobtrusiveness and ease of use of the sensor floor is a major advantage for practical settings, where it could meaningfully support physiotherapeutic diagnostics, and revolutionise the assessment of gait patterns.

## Figures and Tables

**Figure 1 sensors-21-01086-f001:**
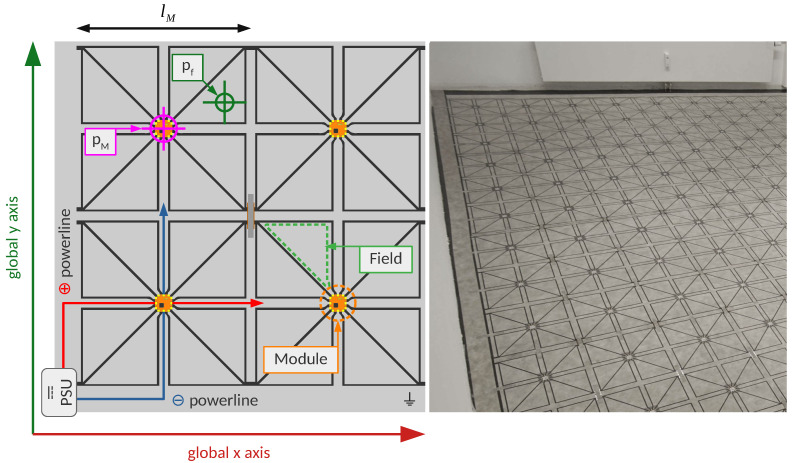
Left side: Schematic of SensFloor. On the bottom left module, the power transmission ways are shown (PSU = Power Supply Unit). The top left module shows the coordinates of a module center (
$p_{M}$
) and a sensor field (
$p_{f}$
). The module length is typically 
$l_{M}=0.38$
 m for Gait Analysis. The electric capacitance is measured on the sensor fields which are shaped as triangles. Right side: SensFloor sensor modules in the study lab. This photo was taken during the installation of the sensors. Afterwards, a carpet was laid on top of the modules, concealing them completely.

**Figure 2 sensors-21-01086-f002:**
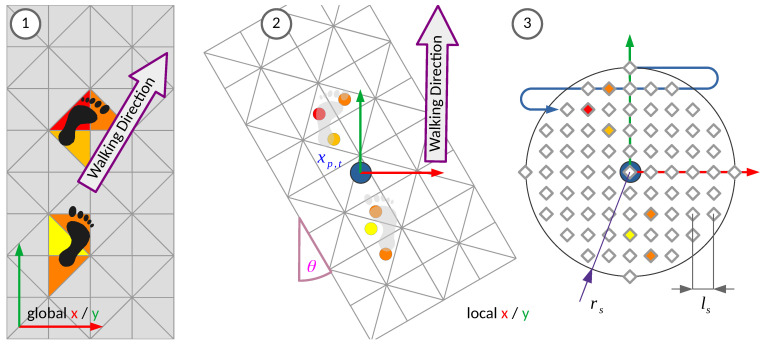
Transformation of the capacitance measurements into the local coordinate system of the tracked person and resampling. 1. Sensor activations in the global coordinates, with examples of how a foot on the floor influences the measured capacitances (grey = unoccupied, yellow to red = level of electric capacitance). 2. The sensor state is translated to the tracked position 
$x_{p,t}$
 and rotated into the walking direction by walking angle 
θ
. 3. The sensor activations are resampled to the grid that is shown with the diamonds, which is defined by the sampling resolution 
$l_{s}$
 and radius 
$r_{s}$
. The sample point that is closest to a transformed sensor field centroid takes its capacitance value. Then, the grid is reshaped into vector form as shown by the meandering blue arrow.

**Figure 3 sensors-21-01086-f003:**
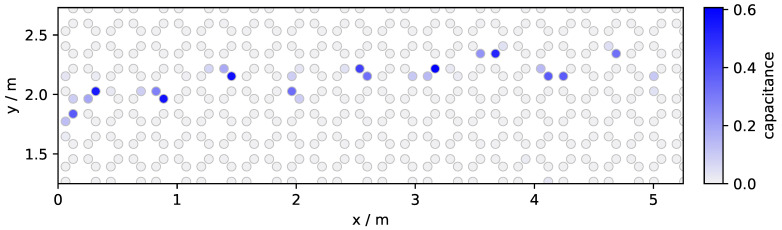
Example data of a person walking on the sensor floor. This plot is generated by showing the maximum capacitance per field that was measured in a single recording. The circles are the centroids of the sensor fields, clusters of centroids with a high maximum capacity are interrelated to steps (footfalls).

**Figure 4 sensors-21-01086-f004:**
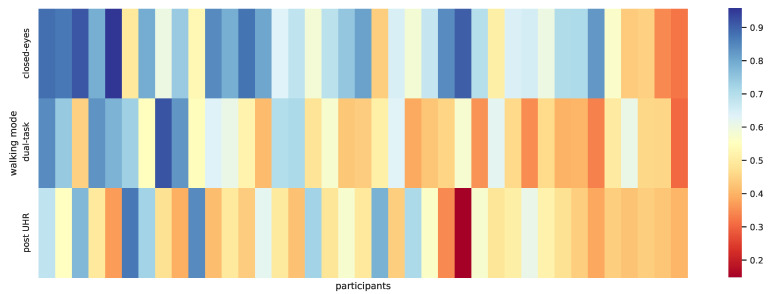
Classification accuracies visualised for the individual participants when classifying between different walking modes. The participants go from left to right, every column is one participant. The walking modes are: normal vs. closed-eyes, normal vs. dualtask, normal vs. post UHR. The columns are sorted from left to right according to the mean accuracy over all three modes for one participant.

**Table 1 sensors-21-01086-t001:** Results for generalised classification.

Mode	Accuracy	Precision	Recall	$F_{1}$ Score
normal vs. closed-eyes	0.77	0.80	0.80	0.80
normal vs. dual-task	0.56	0.58	0.58	0.58
normal vs. post uhr	0.50	0.48	0.46	0.47

## Data Availability

The data presented in this study will be openly available in the future in osf.io at DOI 10.17605/OSF.IO/XMSU2 after further analyses have been conducted by the authors.

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
