# Peer review of "Detecting Walking Challenges in Gait Patterns Using a Capacitive Sensor Floor and Recurrent Neural Networks"

_sensors, 2021, doi:10.3390/s21041086_

Round 1
Reviewer 1 Report
This paper aims to test accuracy of recurrent neural network to classify changes in the gait pattern while walking. The authors defined different walking modes (to walk at their normal pace; faster than normal; sower then normal; with the eyes closed and while performing a dual-task). The classification hypotheses are three: 1) Predicting the walking mode – idiosyncratic; 2) Predicting the walking mode – Generalised; 3) Predicting the UHR repetitions. Characteristic of this paper is also the tool used for acquiring gait, that is a commercial sensing floor usually used in several applications. What is interesting in this paper is exactly the proposal to combine this tool with machine learning analysing directly the time series acquired by the floor.
The paper is well written. A deep description of the SensorFloor is provide. The results obtained are not so impressive, but it is a beginning that encourages future works.
I have only few suggestions:
Line 84 consider these papers that validate Kinect for gait assessment in other pathologies
Summa, S., G. Tartarisco, M. Favetta, A. Buzachis, A. Romano, G. M. Bernava, A. Sancesario et al. "Validation of low-cost system for gait assessment in children with ataxia." Computer Methods and Programs in Biomedicine 196 (2020): 105705.
Grobelny, Anuschka, Janina R. Behrens, Sebastian Mertens, Karen Otte, Sebastian Mansow-Model, Theresa Krüger, Elona Gusho et al. "Maximum walking speed in multiple sclerosis assessed with visual perceptive computing." PLoS One 12, no. 12 (2017): e0189281.
Muñoz, Beatriz, Yor Jaggy Castaño-Pino, Juan David Arango Paredes, and Andrés Navarro. "Automated gait analysis using a Kinect camera and wavelets." In 2018 IEEE 20th International Conference on e-Health Networking, Applications and Services (Healthcom), pp. 1-5. IEEE, 2018.
Ma, Yunru, Kumar Mithraratne, Nichola C. Wilson, Xiangbin Wang, Ye Ma, and Yanxin Zhang. "The Validity and Reliability of a Kinect v2-Based Gait Analysis System for Children with Cerebral Palsy." Sensors 19, no. 7 (2019): 1660.
Line 100: there is a typo
Lines 190-192: Which is the area of the sensing field? What happens if both feet touch the same sensor field?
Line 213: Is there a receiver for all the modules?
Author Response
Thank you for your review and the valuable suggestions on our manuscript „Detecting Walking Challenges in Gait Patterns Using a Capacitive Sensor Floor and Recurrent Neural Networks“. We appreciate the time and effort that you spent on giving feedback and the helpful comments on how to improve the paper.
Everything that we changed is highlighted in the revised version, which you can find in the attachment.
In detail, we have addressed your points as follows:
1. Line 84 consider these papers that validate Kinect for gait assessment in other pathologies
We went through the papers you suggested, and agree that they contribute to emphasizing the validation of the Kinect sensor in the context of gait analysis. We added the references to the bibliography and cited them in paragraph 2.2. at the relevant position (lines 88-89), also giving some more context on how they were used in these research projects.
2. Line 100: there is a typo
The typo at line 100 was corrected, one „of“ was removed (now: line 105).
3. Lines 190-192: Which is the area of the sensing field? What happens if both feet touch the same sensor field?
We added the area values of the sensor fields (which depend on the SensFloor resolution) to the places where we also describe the geometric measures of the sensor fields (lines 209-211). For the case of both feet touching the same sensor field, we added a description that this will increase the surface area and hence the measured capacitance (lines 215-217).
4. Line 213: Is there a receiver for all the modules?
In most cases, there is a single receiver for all modules in one room. We added a statement which describes that this is the case, and also supplemented information about the radio range, which is the technical reason why one receiver per room is indeed enough (lines 225-227).

Reviewer 2 Report
I find the subject to be of great interest, especially in the current medical context.
Gait analysis can be very useful in long term supervision for elderly patients, especially for elder patients suffering either from reduced mobility, either from specific illnesses that have reduced mobility associated with new stages (for example, patients with Parkinson's - who will have a severe mobility reduction and gait modification when their state deteriorates).
Overall I find the article to be well structured, well documented, and based on a proper experiment.
I would recommend expanding the initial explanations on:
- why the authors chose to focus on determining different waking patterns for each individual (versus, for example, distinguishing between individuals);
- why the selected test (versus, for example, having the patient run or dance);
- if this approach could be used to determine if an individual's gait is suffering any modifications.
Author Response
Thank you for the review of our paper „Detecting Walking Challenges in Gait Patterns Using a Capacitive Sensor Floor and Recurrent Neural Networks“, and for the effort and time that you dedicated to giving us valuable feedback and comments. We appreciate the additional questions that you brought up in the review, and revised the manuscript to answer them.
In the attachment, you can find the revised version of the manuscript, with all changes highlighted.
We addressed your questions as follows:
1. why the authors chose to focus on determining different waking patterns for each individual (versus, for example, distinguishing between individuals);
We incorporated this point in two ways:
1.) For clarifying the question of the focus on individuals, we extended the method section with a line about the purpose of the experiment design, generalised and idiosyncratic (lines 377-379).
2.) We agree that distinguishing between individuals is a worthwhile research goal for a future project, that might even be possible with the current dataset. Therefore we added a sentence describing this in the outlook section (lines 494-496).
2. why the selected test (versus, for example, having the patient run or dance);
We think this is an important question, and added answers to this in the introduction (line 45-46). The choice of tests was based on the purpose of generating very subtle variations in the walking patterns, with the idea to evaluate the whole process on a task that is inherently difficult. For more expressive manipulations of the gait patterns it would probably give better results, but serve less as a lower bound of what can be detected.
3. if this approach could be used to determine if an individual's gait is suffering any modifications.
This seems like a very plausible possibility, we made this clearer by adding a sentence in the introduction (line 46-48) as well as in the conclusion (line 515-518) concerning the idea of applying the method to detect modifications that are a result from medical conditions.

Reviewer 3 Report
The paper is a relatively original contribution in the field of walk analysis, deep
process and diagnosis using floor sensors. It is pertinent, well structured, and
relevant for publication with some minor modifications defined in the attached document.

Author Response
Thank you very much for your very helpful and thorough review on our manuscript „Detecting Walking Challenges in Gait Patterns Using a Capacitive Sensor Floor and Recurrent Neural Networks“. We appreciate the time and effort you have dedicated to giving us valuable feedback, suggestions and comments.
We attached a revised version of the manuscript, where the changes are highlighted.
In detail, your points and suggestions were addressed as follows:
1. The title could be: Introduction – General context
This is the only point we did not follow, for now we left the title of the first section as „Introduction“.
2. This section (lines 148-154) could be slightly reinforced by 4-5 additional lines precising the main characteristics of LSTM and –later –MLP (Multi-Layer Perceptron).
We fully agree to this and added paragraphs about the main characteristics for both the LSTM and MLP architectures (lines 158-164, lines 167-172).
3. It could be interesting to make a short «topological» allusion to the advantages of 2D triangular mesh.
We agree and think this point is very important as the triangular shape is somewhat „uncommon“ and therefore should be explained. Also, there are indeed interesting reasons for the triangular shape. We gave an explanation by adding a paragraph to chapter 3.1. describing the reasoning and technological considerations that led to this specific shape (lines 194-201).
4. Why not precising –at this point –a possible analogy with dynamic state-description, possibly in relation with Newmark’s scheme? One or two lines could precise this.
We also agree to this point, as the state update can be interpreted as a dynamical discrete-time system, and already is formalised in a similar way. We added sentences that describe this analogy, and that the SensFloor message would be the input to such a system (lines 263-264).
5. The analysis proposed § 3.4 for the data could be enriched (around line 339) by a short precision (1-2 sentences) about the robustness (and thus, the confidence) reached by the process.
We followed this suggestion by adding the information about the measures that we took in advance to ensure to get a robust Machine Learning model. We very much agree that this is important to add, to emphasize that our experiments are repeatable, and that we took precautions against relying on unstable findings (lines 369-372).
6. Concerning the prediction of Unilateral Heel-Rise (UHR) repetitions, in the last part (§ 4.3) as you mentioned that the true and predicted values did not correlate, you could add that it is a limit of the study.
We added that this is indeed a limit of the study (lines 415-416).
7. § 5.4 envisaging perspectives could be a little enriched. It could be coupled with the conclusion, which is a little short (1 or 2 additional sentences?)
We added a further idea to the envisaging perspectives (line 495-496), concerning the possibility to distinguish between individuals.
Also, we enriched the conclusion with the prospect of how the method opens up future research possibilities in the area of frailty and gait patterns of the elderly (lines 515-518).

Reviewer 4 Report
The paper describes a method of recognising walking gait patterns for human objects using a commercially available sensor mat and analysing the data using recurrent neural networks. the work has clearly explained and the results have been presented well.
Author Response
Thank you very much for taking the time and effort to review our manuscript „Detecting Walking Challenges in Gait Patterns Using a Capacitive Sensor Floor and Recurrent Neural Networks“. We appreciate it a lot and are happy to hear that you assess it to be good for publication. Some changes came up as a result from the other reviews, we want to let you know of this by attaching the revised version where all differences are highlighted.
